# Clinical Outcomes and Risk Factors of Heart Transplantation Patients Experiencing Gastrointestinal Bleeding

**DOI:** 10.3390/biomedicines12081845

**Published:** 2024-08-14

**Authors:** Wangzi Li, Chiyuan Zhang, Xianming Zhou, Qian Xu, Kan Wang, Rong Lin, Jiawei Shi, Nianguo Dong

**Affiliations:** 1Department of Cardiovascular Surgery, Union Hospital, Tongji Medical College, Huazhong University of Science and Technology, 1277 Jiefang Ave, Wuhan 430022, China; 2Department of Cardiovascular Medicine, Xiangya Hospital, Central South University, Xiangya Rd 87, Changsha 410008, China; 3Department of Cardiovascular Surgery, Xiangya Hospital, Central South University, Xiangya Rd 87, Changsha 410008, China; 4Department of Gastroenterology, Union Hospital, Tongji Medical College, Huazhong University of Science and Technology, 1277 Jiefang Ave, Wuhan 430022, China

**Keywords:** heart transplantation, gastrointestinal bleeding, clinical outcomes, risk factors, nomogram

## Abstract

Gastrointestinal bleeding (GIB) after heart transplantation (HT) remains a significant clinical issue. This study aimed to explore the incidence, trends, outcomes, and clinical predictors of GIB in HT patients. Adult patients who underwent HT between 2015 and 2021 at Union Hospital were recruited and divided into two groups based on the presence or absence of postoperative GIB. The primary outcomes were evaluated at follow-up. Independent predictors of GIB after HT were identified using a logistic regression analysis. A nomogram prediction model was constructed according to these independent variables, and the accuracy of the model was assessed using the receiver operating characteristic (ROC) curve and the calibration curve. Among the 461 patients, 40 (8.7%) developed GIB post-HT. HT patients with postoperative GIB exhibited higher in-hospital, 30-day, 90-day, and 1-year mortality (all *p* < 0.05). A multivariate analysis was used to identify age, preoperative warfarin, postoperative continuous renal replacement therapy, and postoperative nasogastric tubes as independent risk factors for GIB following HT. A nomogram prediction model was applied using the four variables. The area under the curve (AUC) of this model was 0.852 (95% CI: 0.787–0.917, *p* < 0.001), and the calibration curve was close to the ideal diagonal line. GIB following HT is associated with a poor clinical prognosis. The constructed nomogram demonstrated a favorable predictive value for GIB.

## 1. Introduction

Heart transplantation (HT) is one of the most significant achievements in modern medicine and has become a standard treatment option for patients with end-stage heart failure [1]. In recent decades, advancements in organ donation and preservation, surgical techniques, immunosuppression, and long-term graft surveillance have significantly improved the clinical outcomes of HT recipients. The median survival is now 10.7 years, with 1-year and 5-year survival rates of 82% and 69%, respectively, contributing to the greater success of HT [2,3]. However, the prolonged survival of HT recipients has also increased the risk of postoperative comorbidities, such as renal dysfunction, embolism, and diabetes, which seriously affect the quality of life of these patients [4,5]. 

Gastrointestinal bleeding (GIB) is a relatively rare complication after most cardiac surgeries (e.g., coronary artery bypass grafting, valve repair or replacement) but is associated with high mortality rates [6]. It was reported that postoperative GIB rates in patients who underwent cardiac surgery ranged from 0.07% to 1.6%, with mortality rates reaching up to 47.6% [6,7]. HT recipients are at a high risk of venous thromboembolism and atrial fibrillation, necessitating anticoagulation therapy, which further increases the bleeding risk [8]. Despite this, the contemporary incidence of postoperative GIB in HT patients and its impact on clinical outcomes remain unclear. Therefore, this study aims to investigate the incidence, trends, and predictors of GIB after HT and to analyze their association with clinical outcomes, providing new evidence for risk stratification and clinical practice in these patients.

## 2. Materials and Methods

### 2.1. Patients and Study Design

This is a retrospective study approved by the Ethics Committee of Tongji College, Huazhong University of Science and Technology (No: IORG0003571). From January 2015 to December 2021, adult patients aged 18 years or older who were admitted to Union Hospital for HT were recruited. Patients were excluded if they met any of the following criteria: (1) experienced GIB within 30 days prior to HT; (2) underwent multi-organ transplantation; (3) died during the procedure; (4) had insufficient clinical data.

### 2.2. Data Collection, Variable Definitions, and Grouping

All the patients included in this study underwent a comprehensive assessment and data collection including demographics, medical history, preoperative laboratory and echocardiography results, preoperative treatment, operative details, and postoperative treatment. Postoperative GIB was defined as the occurrence of GIB symptoms or signs following HT, including melena, hematochezia, hematemesis, or a positive occult blood test (OB) result in feces or gastric juice specimens [9,10]. All the patients were divided into two groups based on postoperative GIB (non-GIB group and GIB group), and were followed up for 1 year. The primary outcomes measured were in-hospital, 30-day, 90-day, and 1-year mortality.

### 2.3. Statistical Analysis

The data are presented as means ± standard deviations (SDs) for normally distributed continuous variables, as medians and interquartile ranges (IQRs) for non-normally distributed continuous variables, and as numbers (n) with percentages (%) for categorical variables. Between-group comparisons of normally distributed values were performed using *t* tests, while non-normally distributed values were analyzed using the Mann–Whitney *U* test. Categorical variables were compared using the Chi-squared test. Trend analyses were conducted using the Cochran–Armitage test. An unadjusted Cox proportional hazards model was used to estimate the cumulative cause-specific hazard of all-cause mortality in association with postoperative GIB after HT, and time-to-event curves were presented as cumulative incidence functions. Univariate and multivariate logistic regression analyses were performed to identify independent factors for GIB (variables with a *p* value of <0.1 and clinical significance were selected for multivariate regression analysis), which were then used for the construction of risk the prediction model as a nomogram. The discriminative ability of the nomogram was assessed using the receiver operating characteristic (ROC) curve and the calibration curve. All statistical analyses were carried out using R Studio (version 4.3.2). A two-sided *p* < 0.05 was considered statistically significant.

## 3. Results

### 3.1. Patient Characteristics

A total of 507 adult patients who underwent HT were initially recruited for this study. A total of 3 patients with GIB within 30 days before HT, 9 patients who underwent multi-organ transplantation, 7 patients who died during the procedure, and 27 patients with insufficient clinical data were excluded. Consequently, 461 adult patients were enrolled in this study (Figure 1), of which 20.6% were male, and the median age was 50 years. The underlying diagnosis of heart failure in the overall population is presented in Figure 2, with dilated cardiomyopathy (DCM) accounting for more than half of HT cases. From 2015 to 2021, the incidence of GIB after HT increased from 3.1% to 20.0%, and the Cochran–Armitage test demonstrated that this trend remained significant (*p*_trend_ < 0.001, Figure 3).

The baseline characteristics are shown in Table 1. Compared to the non-GIB group, the GIB group was older (*p* < 0.05) and had higher activated partial thromboplastin times (APTTs) and international normalized ratio (INR) levels, as well as lower red blood cell (RBC) counts and hemoglobin (Hb) and albumin levels (all *p* < 0.05). However, none of these indicators had clinical significance. Additionally, the preoperative use of warfarin and extracorporeal membrane oxygenation (ECMO) was higher in this group (all *p* < 0.05). In the intraoperative and postoperative data, the GIB group had longer cardiopulmonary bypass (CPB) times, operation times, and intubation times (all *p* < 0.05) and received more continuous renal replacement therapy (CRRT), ECMO, and nasogastric tubes after surgery (all *p* < 0.05).

### 3.2. Postoperative GIB and Survival in HT Patients

As shown in Table 1, the intensive care unit (ICU) and postoperative hospital stay were longer and the rates of in-hospital, 30-day, 90-day, and 1-year mortality were higher in the GIB group (all *p* < 0.05). In addition, the mortality rate was particularly evaluated during the first 90 days post-HT. However, the postoperative GIB patients who survived the initial critical months did not exhibit a significantly worse long-term prognosis (Figure 4).

### 3.3. Risk Factors of Postoperative GIB in HT Patients

To further explore the independent factors associated with GIB after HT, univariate and multivariate logistic regression analyses were performed. According to the results of the univariate analysis, variables with a *p* value of <0.1 and clinical significance were selected for the multivariate regression analysis. After adjusting for a series of variables (e.g., male, cross-clamp time, operation time), age (OR: 1.04, 95% CI: 1.00–1.07, *p*: 0.037), preoperative warfarin (OR: 3.01, 95% CI: 1.07–8.48, *p*: 0.037), postoperative CRRT (OR: 6.27, 95% CI: 2.50–15.72, *p* < 0.001), and postoperative nasogastric tube (OR: 5.73, 95% CI: 2.10–15.66, *p*: 0.001) were significantly associated with postoperative GIB in the patients with HT, as illustrated in Figure 5 and Table 2. 

Based on the results of the multivariate analysis, a nomogram model containing four independent variables was constructed to predict the occurrence of GIB after HT (Figure 6). The prediction model visually emphasized age as the most critical predictor. The probability of postoperative GIB was determined based on the value at a vertical line from the corresponding total points, which was the sum of the relative scores for each parameter. In addition, the predictive accuracy of this model was evaluated based on the ROC curve and the calibration curve. As shown in Figure 7, the area under the curve (AUC) was 0.852 (95% CI: 0.787–0.917, *p* < 0.001), with a sensitivity of 80.0% and specificity of 80.5%, and the calibration curve closely aligned with the ideal diagonal line.

## 4. Discussion

In the current study, we enrolled 461 patients who underwent HT and sought to ascertain the incidence, trends, outcomes, and clinical predictors of postoperative GIB in these patients. The primary observations gleaned from our study are as follows: (1) 8.7% (40/461 HT patients) of the patients developed GIB postoperatively, and its incidence increased between 2015 and 2021; (2) postoperative GIB patients after HT exhibited a higher risk of in-hospital, 30-day, 90-day, and 1-year mortality; (3) age, preoperative warfarin, postoperative CRRT, and postoperative nasogastric tubes were independent risk factors for GIB after HT. Additionally, a nomogram prediction model was established based on these variables, providing a more effective and accurate means of evaluating the possibility of this complication, thereby offering a basis for decision-making in clinical practice.

Over the last 50 years, HT has undergone significant advancements in improving the survival of patients with advanced heart failure. Nonetheless, the persistence of postoperative complications poses a formidable challenge to clinical management, as they are frequently associated with adverse clinical outcomes [11]. It is reported that GIB following cardiac surgery is a relatively rare entity but carries a pronounced mortality risk [6,7]. For instance, a cohort study involving 2956 patients who underwent cardiac surgery (e.g., aortocoronary bypass grafting, valve replacement, aortic aneurysm) revealed the postoperative incidence of GIB to be 0.9%, with an in-hospital mortality rate of 35% [12]. Another study based on three prospectively maintained databases of 9017 cardiac surgery patients (coronary artery bypass grafting and valve procedures) reported an overall incidence of postoperative GIB of 1.01%, with duodenal ulceration identified as the predominant bleeding source, constituting 78% of cases [13]. These GIB patients had prolonged postoperative hospital stays and a heightened 30-day mortality rate (8.8%) compared to the control group [13]. However, despite the high risk of bleeding complications among HT recipients [8], the currently available research lacks a comprehensive investigation into the postoperative GIB in this cohort. In our study, we retrospectively analyzed a cohort of 461 HT patients and found that 8.7% experienced postoperative GIB, with the incidence increasing from 3.1% in 2015 to 20.0% in 2021.

To the best of our knowledge, this is the first study to focus on postoperative GIB among HT patients. The higher incidence of GIB observed in the patients with HT compared to those who had undergone other forms of cardiac surgery may be attributed to several factors that are intrinsic to the transplantation procedure and the perioperative management of these patients, including pre-existing comorbidities, immunosuppressive therapy [14], anticoagulant and antiplatelet therapy [15], altered hemodynamics [16], and reperfusion injury [17]. Additionally, we observed that these postoperative GIB patients had poorer clinical prognoses, including higher in-hospital (15.0%), 30-day (27.5%), 90-day (50.0%), and 1-year (52.5%) mortality rates and longer postoperative hospital stays during follow-up, consistent with previous reports of GIB following other cardiac surgeries [6]. The patients who survived beyond this initial phase did not show a significantly worse long-term prognosis compared to those without GIB, suggesting that while GIB presents a substantial early risk, its impact may diminish over time if patients overcome the initial critical period. Overall, these findings highlighting the importance of comprehensive risk assessment for developing GIB after HT and suggest the need for vigilant monitoring and tailored management strategies to mitigate the risk of bleeding complications and optimize patient outcomes in the post-transplantation period.

Age emerged as a prominent independent risk factor in our study. The correlation between age and postoperative GIB has been investigated widely. Kim et al. conducted a nationwide population-based study of 1,319,807 patients who underwent various surgeries in Korea and elucidated a significant association between advancing age and the occurrence of postoperative GIB. Specifically, patients aged ≥70 years had an approximately 20-fold increase in risk compared to their counterparts in their 20s [18]. Similarly, Hsu et al. developed a machine learning algorithm to prognosticate postoperative GIB among 159,959 individuals undergoing bariatric surgery, wherein age was one of the five most-influential predictors within the model [19]. Consistent with these findings, our study showed that age was higher in the GIB group (54.50 [47.50–58.00] vs. 50.00 [39.00–57.00] years, *p* = 0.024) and was independently associated with GIB after HT based on a multivariate analysis, which may be due to the high prevalence of various comorbidities in the elderly population [20].

Among a series of preoperative variables, we also observed a significant association between preoperative warfarin and postoperative GIB in the HT recipients. Warfarin, a vitamin K antagonist (VKA), has traditionally served as a primary oral anticoagulant for the prevention of ischemic stroke or systemic embolism [21], particularly in patients with cardiac mechanical valve replacements [22]. However, its clinical utility is tempered by its propensity for hemorrhagic complications [21,23]. In our current study, we analyzed preoperative antithrombotic drugs in two groups and demonstrated that most drugs, including heparin, low-molecular-weight heparin (LMWH), aspirin, and clopidogrel were not related to GIB after HT, except for warfarin. These results suggest that for patients requiring HT after a mechanical valve replacement, greater attention should be paid to their anticoagulant usage prior to the transplant procedure. For instance, patients with mechanical heart valves can be bridged with unfractionated heparin or LMWH until a therapeutic INR has been attained before the waiting period for the transplant procedure.

Furthermore, postoperative CRRT emerged as another important clinical predictor for postoperative GIB, consistent with previous studies [24,25,26]. For instance, Elizabeth Parsons et al. evaluated the outcomes of CRRT in pediatric liver transplantation recipients, revealing an elevated incidence of GIB in patients necessitating CRRT, though the results require further validation through larger-scale studies [25]. Similarly, both Granholm et al. and Asleh et al. confirmed the association between CRRT utilization and heightened GIB risk in patients in the ICU or those with continuous-flow left ventricular assist device (CF-LVAD) implantation, respectively [24,26]. Since most patients undergoing CRRT have preexisting renal disease or acute renal failure, they often experience severe systemic congestion, which in turn increases venous pressure in the mesenteric circulation, leading to elevated shear stress and a higher risk of GIB [27]. Moreover, the concomitant use of anticoagulants during CRRT further amplifies the risk of bleeding [28]. Thus, postoperative CRRT in HT patients should be comprehensively evaluated and individualized.

Interestingly, our analysis revealed that postoperative nasogastric tube use significantly increased the risk of GIB following HT. Nasogastric tubes are commonly used for stomach decompression and the administration of drugs and nutrients [29]. They are often used postoperatively to manage feeding intolerance, nausea, and vomiting after surgery, serving as a standard measure to minimize gastric symptoms and alleviate gastric distension [30,31,32]. Recent debates have questioned the necessity and efficacy of postoperative gastrointestinal tubes and highlighted the potential complications they may cause. Some researchers have reported that postoperative gastrointestinal tube use is associated with the incidence of respiratory complications, gastrointestinal complications, and postoperative pain and discomfort [33,34,35,36]. Our study is the first to demonstrate that postoperative nasogastric tube use increases the risk of GIB following HT, indicating that more mechanical injury, pressure, and irritation may have occurred in the gastrointestinal tract due to gastrointestinal tube insertion in patients after HT, leading to erosions, ulcerations, and ultimately bleeding. This finding suggests the need for clinicians to further discuss the indications for postoperative nasogastric tube use in such patients, exercise greater caution when using gastrointestinal tubes in HT patients, and closely monitor for signs of GIB in those with indwelling tubes.

Notably, several intraoperative indicators, including CPB time, cross-clamp time, and operation time, have been reported as risk factors for gastrointestinal complications [10]. However, evidence of their association with postoperative GIB in HT patients remains inconclusive. Previous studies have suggested that prolonged CPB and aortic cross-clamp times may exert adverse effects on abdominal perfusion and splanchnic perfusion during CPB procedures, leading to inadequate metabolic supply and exacerbating gastrointestinal complications [10,37]. However, most of the evidence primarily focuses on intestinal ischemic injury rather than specifically on GIB. In our study, the univariate analyses revealed that the CPB time, aortic cross-clamp time, and operation time were significant, but after adjusting for other clinical variables, their significance diminished, suggesting that these intraoperative variables may not independently predispose individuals to GIB.

Several limitations should be considered in the current study. First, due to the retrospective study design, selection bias and some residual confounders cannot be ruled out, despite the use of a multivariate analysis for the adjustment of relative variables. Second, there is a lack of assessment of the severity and cause of the postoperative GIB in the HT patients, which may also impact the research results. Third, the data analysis is based on a single center, so the external validity of our findings should be further evaluated. However, our findings have important implications for the management of HT patients. Recognizing high-risk patients with HT using our predictive model allows for targeted interventions and closer monitoring, potentially mitigating the incidence and severity of GIB. For instance, optimizing anticoagulation therapy and closely monitoring coagulation parameters preoperatively could reduce the risk of postoperative GIB after HT. Additionally, minimizing CRRT time and ensuring vigilant postoperative care could further improve the outcomes of these patients.

## 5. Conclusions

In summary, our study demonstrated that GIB is a significant complication following HT, associated with increased mortality and morbidity, particularly in the early postoperative period. Our nomogram predictive model incorporating its independent risk factors, including age, preoperative warfarin, postoperative CRRT and postoperative nasogastric tubes, showed favorable predictive value.

## Figures and Tables

**Figure 1 biomedicines-12-01845-f001:**
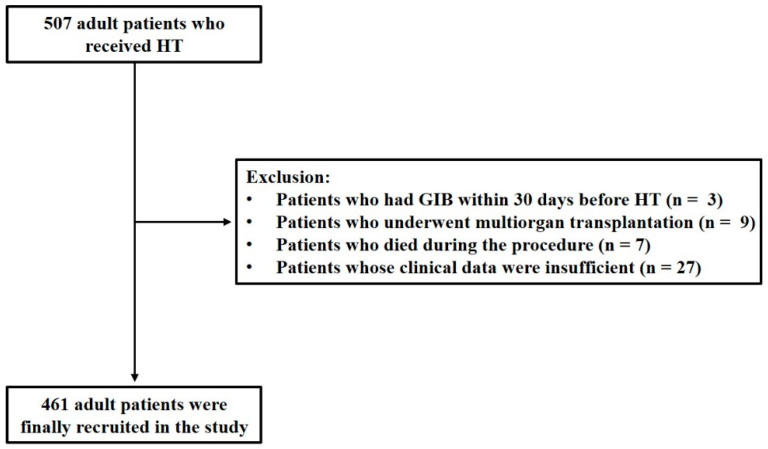
Flowchart of the retrospective study. HT: heart transplantation; GIB: gastrointestinal bleeding.

**Figure 2 biomedicines-12-01845-f002:**
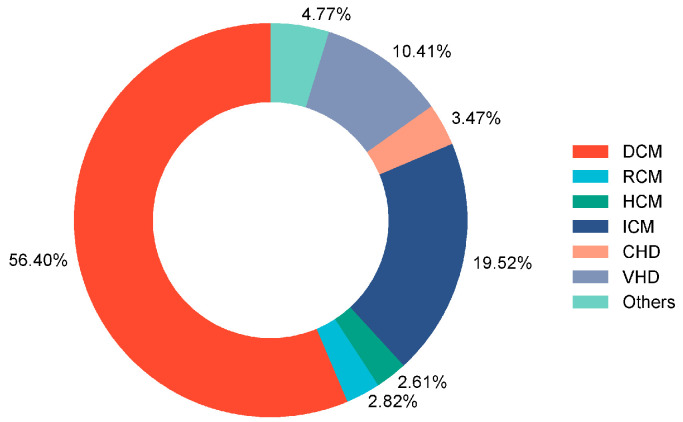
Underlying diagnosis of heart failure resulting in HT among enrolled patients. DCM: dilated cardiomyopathy; RCM: restrictive cardiomyopathy; HCM: hypertrophic cardiomyopathy; ICM: ischemic cardiomyopathy; CHD: congenital heart disease; VHD: valvular heart disease; HT: heart transplantation.

**Figure 3 biomedicines-12-01845-f003:**
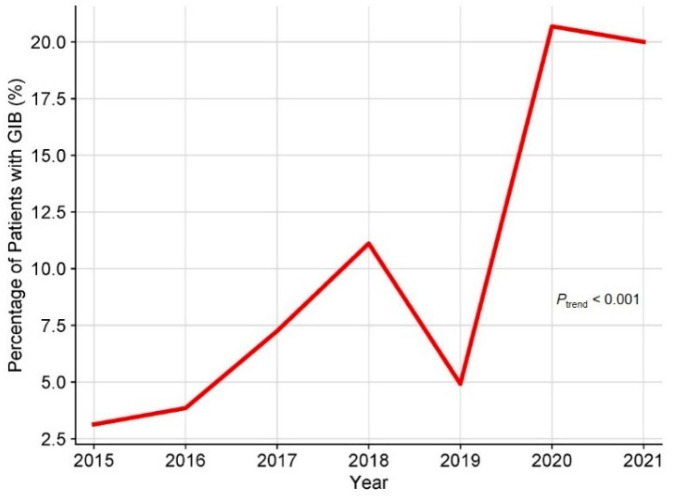
Time trend of crude incidence of GIB after HT. GIB: gastrointestinal bleeding; HT: heart transplantation.

**Figure 4 biomedicines-12-01845-f004:**
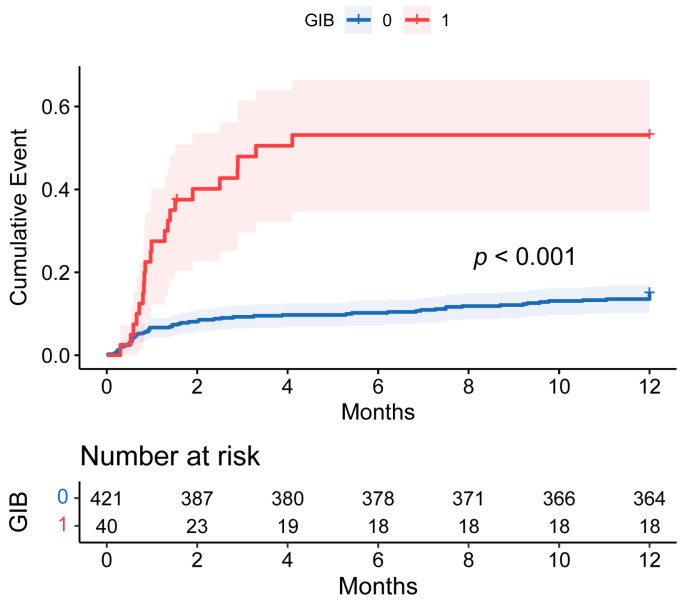
Cumulative incidence of events comparing 1-year mortality between patients with and without postoperative GIB after HT. The shaded area around the curves represents the 95% CI. GIB: gastrointestinal bleeding; HT: heart transplantation; CI: confidence interval.

**Figure 5 biomedicines-12-01845-f005:**
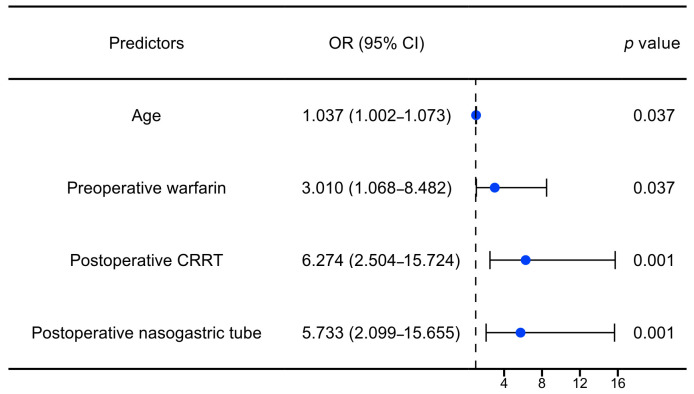
Forest plot of independent risk factors for GIB after HT. OR: odds ratio; CI: confidence interval; CRRT: continuous renal replacement therapy.

**Figure 6 biomedicines-12-01845-f006:**
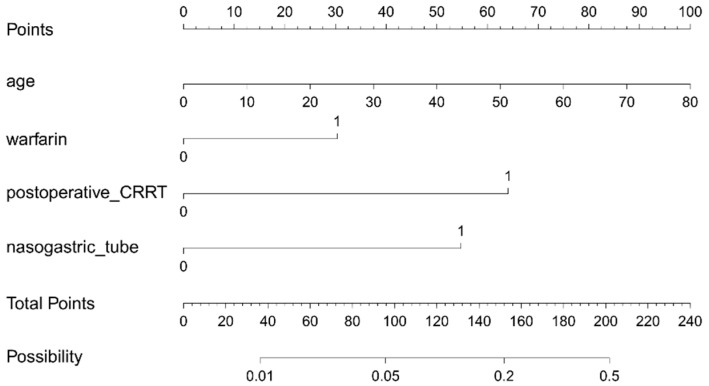
Nomogram for the prediction of postoperative GIB risk in HT patients. CRRT: continuous renal replacement therapy.

**Figure 7 biomedicines-12-01845-f007:**
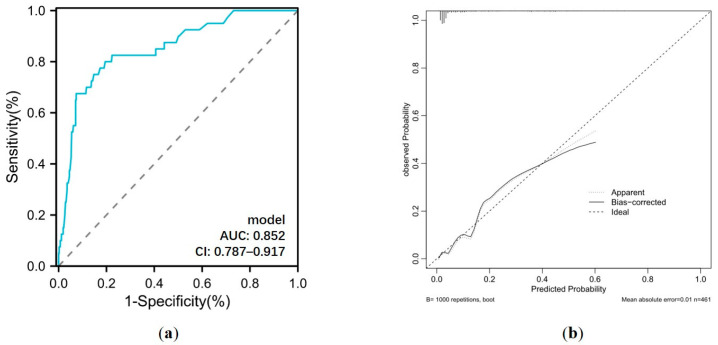
(**a**) ROC curve and (**b**) calibration curve for postoperative GIB prediction model. AUC: area under the curve; CI: confidence interval.

**Table 1 biomedicines-12-01845-t001:** Baseline characteristics of the study patients.

	Non-GIB (n = 421)	GIB (n = 40)	*p* Values
Age (years)	50.00 (39.00–57.00)	54.50 (47.50–58.00)	0.024
Gender, male (%)	84 (20.0)	11 (27.5)	0.259
BMI (kg/m^2^)	23.02 ± 3.97	22.02 ± 2.89	0.120
Blood type			0.774
A, n (%)	146 (34.7)	12 (30.0)	
B, n (%)	116 (27.6)	11 (27.5)	
AB, n (%)	31 (7.4)	2 (5.0)	
O, n (%)	128 (30.4)	15 (37.5)	
Current smoking, n (%)	174 (41.3)	12 (30.0)	0.163
Current drinking, n (%)	116 (27.6)	7 (17.5)	0.170
Medical history			
Hypertension, n (%)	70 (16.6)	3 (7.5)	0.131
Hyperlipemia, n (%)	28 (6.7)	0 (0.0)	0.157
Diabetes, n (%)	85 (20.2)	8 (20.0)	0.977
Gastrointestinal disease, n (%)	48 (11.4)	5 (12.5)	0.796
NYHA classification			0.932
II, n (%)	1 (0.2)	0 (0.0)	
III, n (%)	18 (4.3)	2 (5.0)	
IV, n (%)	402 (95.5)	38 (95.0)	
Cardiac operation, n (%)	121 (28.7)	17 (42.5)	0.069
Dialysis, n (%)	4 (1.0)	0 (0.0)	1.000
Preoperative data			
RBC (×10^12^/L)	4.49 (4.08–4.89)	4.16 (3.72–4.69)	0.004
Hb (g/L)	138.00 (122.00–149.00)	127.50 (105.75–141.25)	0.004
WBC (×10^9^/L)	6.36 (4.96–8.00)	5.97 (4.46–6.93)	0.089
PLT (×10^9^/L)	172.50 (138.25–221.75)	173.00 (119.25–235.25)	0.782
albumin (g/L)	39.50 (36.60–42.30)	37.90 (34.90–40.30)	0.025
AST (mmol/L)	28.00 (21.00–41.00)	26.00 (18.25–38.00)	0.737
ALT (mmol/L)	28.00 (18.00–46.00)	23.50 (13.25–44.75)	0.226
TBIL (mmol/L)	22.00 (14.20–33.50)	27.05 (13.68–40.85)	0.737
Cr (μmol/L)	88.65 (72.43–108.33)	87.35 (72.65–119.78)	0.996
BUN (mmol/L)	7.31 (5.84–9.57)	7.55 (6.30–11.09)	0.291
TG (mmol/L)	1.06 (0.78–1.43)	0.95 (0.74–1.29)	0.381
LDL-C (mmol/L)	2.14 (1.69–2.63)	2.09 (1.61–2.76)	0.876
APTT (s)	39.00 (36.20–43.25)	40.60 (38.18–46.50)	0.027
INR	1.17 (1.07–1.37)	1.31 (1.16–1.60)	0.003
NT-proBNP (pg/mL)	3079.15 (1471.25–6723.78)	3930 (1310–7974)	0.597
LVEF (%)	24.20 (20.00–29.00)	22.20 (16.00–29.50)	0.264
Preoperative treatment			
Warfarin, n (%)	45 (10.7)	10 (25.0)	0.017
Heparin, n (%)	97 (23.0)	11 (27.5)	0.525
LMWH, n (%)	108 (25.7)	8 (20.0)	0.431
Aspirin, n (%)	77 (18.3)	10 (25.0)	0.300
Clopidogrel, n (%)	21 (5.0)	3 (7.5)	0.453
Intubation, n (%)	6 (1.4)	1 (2.5)	0.473
IABP, n (%)	6 (1.4)	2 (5.0)	0.147
ECMO, n (%)	7 (1.7)	3 (7.5)	0.047
Intraoperative data			
Pulmonary artery systolic pressure (mmHg)	48.00 (34.00–61.25)	50.00 (40.00–59.00)	0.557
Cross clamp time (min)	30.00 (26.00–36.00)	33.00 (26.00–43.50)	0.087
CPB time (min)	106.00 (91.00–130.00)	133.00 (103.50–154.75)	<0.001
Operation time (min)	240.00 (210.00–300.00)	300.00 (242.50–360.00)	<0.001
Postoperative data			
First 24 h of drainage (mL)	350.00 (250.00–520.00)	375.00 (152.50–530.00)	0.700
Intubation time (min)	2160.00 (1401.25–3254.25)	6210.00 (2510.25–16,530.00)	<0.001
CRRT, n (%)	48 (11.4)	26 (65.0)	<0.001
IABP, n (%)	161 (38.2)	21 (52.5)	0.078
ECMO, n (%)	17 (4.0)	9 (22.5)	<0.001
Nasogastric tube, n (%)	97 (23.0)	31 (77.5)	<0.001
ICU stay (hours)	214.00 (158.50–279.50)	424.00 (196.00–714.00)	<0.001
Postoperative hospital stay (days)	34.00 (26.00–46.00)	49.00 (29.25–72.75)	0.001
In-hospital deaths	20 (4.8)	6 (15.0)	0.018
30-day deaths	28 (6.7)	11 (27.5)	<0.001
90-day deaths	39 (9.3)	20 (50.0)	<0.001
1-year deaths	63 (15.0)	21 (52.5)	<0.001

Data are expressed as means ± standard deviations, n (%), or medians (interquartile ranges). GIB: gastrointestinal bleeding; BMI: body mass index; NYHA: New York Heart Association; RBC: red blood cell; Hb: hemoglobin; WBC: white blood cell; PLT: platelet; AST: aspartate transaminase; ALT: alanine aminotransferase; TBIL: total bilirubin; Cr: creatinine; BUN: blood urea nitrogen; TG: triglyceride; LDL-C: low-density lipoprotein cholesterol; APTT: activated partial thromboplastin time; INR: international normalized ratio; NT-proBNP: N-terminal pro-B-type natriuretic peptide; LVEF: left ventricular ejection fraction; LMWH: low-molecular-weight heparin; IABP: intra-aortic balloon pump; ECMO: extracorporeal membrane oxygenation; CPB: cardiopulmonary bypass; CRRT: continuous renal replacement therapy; ICU: intensive care unit.

**Table 2 biomedicines-12-01845-t002:** Univariate and multivariate logistic regression analysis of postoperative GIB in patients with HT.

	Univariate Analysis		Multivariate Analysis	
	OR (95% CI)	*p* Values	OR (95% CI)	*p* Values
Age	1.04 (1.00–1.07)	0.027	1.04 (1.00–1.07)	0.037
Male	1.52 (0.73–3.17)	0.262	1.11 (0.45–2.77)	0.820
BMI	0.93 (0.85–1.02)	0.119		
Cardiac operation	1.92 (0.93–3.98)	0.078		
Gastrointestinal disease	1.11 (0.42–2.97)	0.835		
RBC	0.62 (0.39–0.99)	0.046		
Hb	0.98 (0.96–0.99)	0.003		
WBC	0.89 (0.77–1.03)	0.128		
albumin	0.93 (0.87–1.00)	0.050		
APTT	1.01 (0.99–1.04)	0.244		
INR	1.24 (0.89–1.73)	0.197		
Preoperative warfarin	2.79 (1.28–6.07)	0.010	3.01 (1.07–8.48)	0.037
Preoperative ECMO	4.80 (1.19–19.32)	0.027	1.741 (0.34–8.94)	0.507
Cross-clamp time	1.03 (1.00–1.06)	0.056	1.00 (0.96–1.04)	0.931
CPB time	1.00 (1.00–1.01)	0.085	1.00 (0.99–1.01)	0.937
Operation time	1.01 (1.00–1.01)	0.001	1.00 (0.99–1.00)	0.362
Postoperative CRRT	14.43 (7.05–29.53)	<0.001	6.27 (2.50–15.72)	<0.001
Postoperative IABP	1.79 (0.93–3.42)	0.081	0.491 (0.21–1.18)	0.112
Postoperative ECMO	6.90 (2.84–16.75)	<0.001	2.36 (0.70–7.93)	0.164
Postoperative nasogastric tube	11.51 (5.30–25.00)	<0.001	5.73 (2.10–15.66)	0.001

Multivariate analysis adjusted for male, preoperative ECMO, cross-clamp time, CPB time, operation time, postoperative IABP, postoperative ECMO. GIB: gastrointestinal bleeding; HT: heart transplantation; OR: odds ratio; CI: confidence interval; BMI: body mass index; RBC: red blood cell; Hb: hemoglobin; WBC: white blood cell; APTT: activated partial thromboplastin time; INR: international normalized ratio; ECMO: extracorporeal membrane oxygenation; CPB: cardiopulmonary bypass; CRRT: continuous renal replacement therapy; IABP: intra-aortic balloon pump; ICU: intensive care unit.

## Data Availability

The original data are available from the corresponding author upon request. The data are not publicly available due to patient privacy.

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
