# Peer review of "Clinical Outcomes and Risk Factors of Heart Transplantation Patients Experiencing Gastrointestinal Bleeding"

_biomedicines, 2024, doi:10.3390/biomedicines12081845_

Round 1

Reviewer 1 Report

Comments and Suggestions for Authors

This is a potentially interesting paper on the incidence and clinical significance of GI bleeding following heart transplantation.However, several issues should be addressed.

Comments/queries

1. Statements in the introduction based on references 8 and 9 should be removed as they are not related to heart transplantation.

2. How many patients were treated with PPIs, which reduce GI bleeding risk following heart transplantation.

3. Established risk factors for upper GI bleeding following cardiac surgery include advanced age, mechanical ventilation, prolonged elevated international normalized ratio (INR), and cardiopulmonary bypass duration. What was the indication to use a nasogastric tube  in this study? What do the authors recommend regarding the use of nasogastric tubes following heart transplantation? Association does not mean causation.

4. LVAD were not used as a bridge to heart transplantation in this study. How does this fact affect the generalization of study findings?

Comments on the Quality of English Language

Moderate Editing 

Author Response

Reviewer 1: This is a potentially interesting paper on the incidence and clinical significance of GI bleeding following heart transplantation. However, several issues should be addressed.

Comments/queries

Comments 1: Statements in the introduction based on references 8 and 9 should be removed as they are not related to heart transplantation.

Response 1: Thank you for your valuable comments. We deleted the relevant statements of LVADs in the introduction and also removed references 8 and 9.

Comments 2: How many patients were treated with PPIs, which reduce GI bleeding risk following heart transplantation.

Response 2: Thank you for your important comments and constructive suggestions. In our center, HT patients routinely received proton pump inhibitor (PPI) therapy after surgery to prevent gastrointestinal bleeding (GIB). Therefore, 461 patients in our study were treated with PPI, but 40 of them still experienced postoperative GIB. For these patients, we adjusted the dosage of PPI and added somatostatin (n = 11), pituitrin (n = 2), vitamin K1 (n = 12) or provided endoscopy (n = 2) for hemostasis. Unfortunately, compared with the non-GIB patients, the GIB patients had a poorer clinical prognosis including the intensive care unit (ICU) (424.00 [196.00-714.00] vs. 214.00 [158.50-279.50], P < 0.001) and postoperative hospital stays (49.00 [29.25-72.75] vs. 34.00 [26.00-46.00], P = 0.001), and the rates of in-hospital (15.0% vs. 4.8%, P = 0.018), 30-day (27.5% vs. 6.7%, P < 0.001), 90-day (50.0% vs. 9.3%, P < 0.001) and 1-year mortality (52.5% vs. 15.0%, P < 0.001), which were also shown in the results (Table 1). Since the number of patients with GIB after HT in our study was relatively small, the results still need to be further verified in a large sample.

Comments 3: Established risk factors for upper GI bleeding following cardiac surgery include advanced age, mechanical ventilation, prolonged elevated international normalized ratio (INR), and cardiopulmonary bypass duration. What was the indication to use a nasogastric tube in this study? What do the authors recommend regarding the use of nasogastric tubes following heart transplantation? Association does not mean causation.

Response 3: We are very grateful for your valuable suggestion. In our study, nasogastric tubes were used for postoperative nausea and vomiting or administration of feeds and medications for patients unable tolerate oral intake. Currently, although perioperative use of nasogastric tubes in surgical patients is widely accepted[1, 2], the evidence for its benefit in cardiac surgery patients is insufficient[3]. Our findings showed that postoperative nasogastric tube use was an independent risk factor for GIB following HT. This result suggests that the indications for nasogastric tube use in patients after HT should be further discussed and clinicians should exercise greater caution to HT patients with nasogastric tubes, and closely monitor for signs of GIB in those with indwelling tubes. We also added the suggestions to the discussion (Line 292-295).

Comments 4: LVAD were not used as a bridge to heart transplantation in this study. How does this fact affect the generalization of study findings?

Response 4: Thank you for your great point. Nowadays, LVADs have been utilized as either a bridge-to-transplant (BTT) or as destination therapy (DT) for patents with end-stage heart failure[4], but these devices are only available to Chinese patients in recent years[5, 6]. Since the data collected in this study were from 2015 to 2021, and our department started LVADs implantation in 2021, there is currently limited data available for this part. In addition, The Society of Thoracic Surgeons Intermacs 2020 Annual Report showed an increase in DT LVADs to 73.1% and a decrease in BTT LVADs to 8.9% in 2019, indicating that a minority of LVADs were used as the pretransplant setting[7]. Besides, more and more observation studies revealed that LVADs were associated with significant number of adverse events including GIB, infection, thrombosis, stroke and right ventricular failure[4, 7-9]. Our current study focused more on exploring the risk factors of GIB following HT without implantation of LVADs, which could provide some new evidence for other medical centers in China or in developing countries that have not used LVADs as a BTT in HT.

References

  1. Bull A, Pucher PH, Maynard N, et al. Nasogastric tube drainage and pyloric intervention after oesophageal resection: UK practice variation and effect on outcomes. European journal of surgical oncology : the journal of the European Society of Surgical Oncology and the British Association of Surgical Oncology. 2022;48(5):1033-8.
  2. Okabe K, Kaneko R, Kawai T, et al. Efficacy of semi-solidification of enteral nutrients for postoperative nutritional management with a nasogastric tube. Nagoya journal of medical science. 2022;84(2):366-73.
  3. Paleczny S, Fatima R, Amador Y, El Diasty M. Should nasogastric tube be used routinely in patients undergoing cardiac surgery? A narrative review. Journal of cardiac surgery. 2022;37(12):5300-6.
  4. McNamara N, Narroway H, Williams M, et al. Contemporary outcomes of continuous-flow left ventricular assist devices-a systematic review. Annals of cardiothoracic surgery. 2021;10(2):186-208.
  5. Wang X, Zhou X, Chen H, et al. Long-term outcomes of a novel fully magnetically levitated ventricular assist device for the treatment of advanced heart failure in China. The Journal of heart and lung transplantation : the official publication of the International Society for Heart Transplantation. 2024.
  6. Sun YF, Wang ZW, Zhang J, et al. Current Status of and Opinions on Heart Transplantation in China. Current medical science. 2021;41(5):841-6.
  7. Molina EJ, Shah P, Kiernan MS, et al. The Society of Thoracic Surgeons Intermacs 2020 Annual Report. The Annals of thoracic surgery. 2021;111(3):778-92.
  8. Carlson LA, Maynes EJ, Choi JH, et al. Characteristics and outcomes of gastrointestinal bleeding in patients with continuous-flow left ventricular assist devices: A systematic review. Artificial organs. 2020;44(11):1150-61.
  9. Berg D, Lebovics E, Kai M, Spielvogel D. The Predicament of Gastrointestinal Bleeding in Patients With a Continuous-Flow Left Ventricular Assist Device: Pathophysiology, Evaluation, and Management. Cardiology in review. 2019;27(5):222-9.

Reviewer 2 Report

Comments and Suggestions for Authors

Dear authors,

Thank you very much for the opportunity to review the manuscript  Clinical outcomes and risk factors of heart transplantation patients experiencing gastrointestinal bleeding.

Below are comments concerning the article:

In the abstract, please simplify the sentence: The constructed nomogram, incorporating its independent risk factors, demonstrated a favorable predictive value for GIB. Did you analyze the patients with LVAD? There is a group prone to GI bleeding.

Author Response

Reviewer 2:

Dear authors,

Thank you very much for the opportunity to review the manuscript Clinical outcomes and risk factors of heart transplantation patients experiencing gastrointestinal bleeding.

Below are comments concerning the article:

Comments 1: In the abstract, please simplify the sentence: The constructed nomogram, incorporating its independent risk factors, demonstrated a favorable predictive value for GIB. Did you analyze the patients with LVAD? There is a group prone to GI bleeding.

Response 1: We gratefully appreciate your rigorous consideration. Initially, we simplified the sentence you mentioned in the abstract. Additionally, LVADs were an important treatment option for patients with advanced heart failure[4], but this therapy has not been widely used in China[5, 6]. Our center used LVADs as BTT in HT patients since 2021, however, HT patients recruited in our study were from 2015 to 2021. So, we are very sorry that there is limited data available for this part.

References

  1. Bull A, Pucher PH, Maynard N, et al. Nasogastric tube drainage and pyloric intervention after oesophageal resection: UK practice variation and effect on outcomes. European journal of surgical oncology : the journal of the European Society of Surgical Oncology and the British Association of Surgical Oncology. 2022;48(5):1033-8.
  2. Okabe K, Kaneko R, Kawai T, et al. Efficacy of semi-solidification of enteral nutrients for postoperative nutritional management with a nasogastric tube. Nagoya journal of medical science. 2022;84(2):366-73.
  3. Paleczny S, Fatima R, Amador Y, El Diasty M. Should nasogastric tube be used routinely in patients undergoing cardiac surgery? A narrative review. Journal of cardiac surgery. 2022;37(12):5300-6.
  4. McNamara N, Narroway H, Williams M, et al. Contemporary outcomes of continuous-flow left ventricular assist devices-a systematic review. Annals of cardiothoracic surgery. 2021;10(2):186-208.
  5. Wang X, Zhou X, Chen H, et al. Long-term outcomes of a novel fully magnetically levitated ventricular assist device for the treatment of advanced heart failure in China. The Journal of heart and lung transplantation : the official publication of the International Society for Heart Transplantation. 2024.
  6. Sun YF, Wang ZW, Zhang J, et al. Current Status of and Opinions on Heart Transplantation in China. Current medical science. 2021;41(5):841-6.
  7. Molina EJ, Shah P, Kiernan MS, et al. The Society of Thoracic Surgeons Intermacs 2020 Annual Report. The Annals of thoracic surgery. 2021;111(3):778-92.
  8. Carlson LA, Maynes EJ, Choi JH, et al. Characteristics and outcomes of gastrointestinal bleeding in patients with continuous-flow left ventricular assist devices: A systematic review. Artificial organs. 2020;44(11):1150-61.
  9. Berg D, Lebovics E, Kai M, Spielvogel D. The Predicament of Gastrointestinal Bleeding in Patients With a Continuous-Flow Left Ventricular Assist Device: Pathophysiology, Evaluation, and Management. Cardiology in review. 2019;27(5):222-9.

Round 2

Reviewer 1 Report

Comments and Suggestions for Authors

No further comments

Comments on the Quality of English Language

Minor editing